# Analysis of Factors Relevant to Revenue Improvement in Ventral Hernia Repair, Their Influence on Surgical Training, and Development of Predictive Models: An Economic Evaluation

**DOI:** 10.3390/healthcare9091226

**Published:** 2021-09-17

**Authors:** Anas Taha, Bassey Enodien, Vincent Ochs, Marta Bachmann, Maike Gripp, Michel Adamina, Stephanie Taha-Mehlitz, Daniel Frey

**Affiliations:** 1CRAIS, Department of Biomedical Engineering, Faculty of Medicine, University of Basel, 4123 Allschwil, Switzerland; vincent.ochs@unibas.ch (V.O.); michel.adamina@gmail.com (M.A.); 2Department of Visceral and Thoracic Surgery Cantonal Hospital Winterthur, 8400 Winterthur, Switzerland; 3Department of Surgery, GZO Hospital, 8620 Wetzikon, Switzerland; bassey.enodien@gzo.ch (B.E.); marta.bachmann@gzo.ch (M.B.); maike.gripp@gzo.ch (M.G.); daniel.frey@gzo.ch (D.F.); 4Roche Innovation Center Basel, Department of Pharma Research & Early Development, 4070 Basel, Switzerland; 5Clarunis, University Center for Gastrointestinal and Liver Diseases, 4002 Basel, Switzerland; stephanie.taha@clarunis.ch; 6Faculty of Medicine, University of Basel, 4001 Basel, Switzerland

**Keywords:** costs, economy, ventral hernia, incisional hernia, predictive model

## Abstract

Background: Ventral hernia repairs (VHR) are frequent but loss- making. This study aims to identify epidemiological and procedure related factors in VHR and their influence on surgical training. Methods: Data from 86 consecutive patients who underwent VHR in 2019 was collected. Moreover, 66 primary ventral hernias and 20 incisional hernias were repaired in open procedures. Linear regression models were made. Results: Primary VHR procedures showed a mean deficit of −378.17 CHF per case. Incisional hernia repair procedures resulted in a deficit of −1442.50 CHF per case. The two hernia groups were heterogeneous. For the primary VHR procedures, the surgery time (β = 0.564, *p* < 0.001) had the greatest influence, followed by the costs of the mesh (β = −0.215, *p* < 0.001). The epidemiological factors gender (β = 0.143, *p* < 0.01) and body mass index (BMI) (β = −0.087, *p* = 0.074) were also influential. For incisional hernia procedures a surgeon’s experience had the most significant influence (β = 0.942, *p* < 0.001), and the second largest influence was the price of the mesh (β = −0.500, *p* < 0.001). The epidemiological factor BMI (β = −0.590, *p* < 0.001), gender (β = −0.113, *p* = 0.055) and age (β = −0.026, *p* < 0.050) also had a significant influence. Conclusion: Our analysis shows a way of improving financial results in the field of ventral hernia repair. Costs can be visualized and reduced to optimize revenue enhancement in surgical departments. In our analysis primary ventral hernias are an appropriate training operation, in which the experience of the surgeon has no significant impact on costs. In primary VHR procedures, revenue enhancement is limited when using an expensive mesh. However, the treatment of incisional hernias is recommended by specialists. The financial burden is significantly higher with less experience. Therefore, these operations are not suitable for surgical training. The re-operation rate decreases with increasing experience of the surgeon. This directly affects the Patient Related Outcome (PROM) and quality of treatment. Therefore, high-quality training must be enforced. Since financial pressure on hospitals is increasing further, it is crucial to investigate cost influencing factors. The majority of Swiss public hospitals will no longer be able to operate ventral hernias profitably without new concepts. In addition to purchasing management, new construction projects, and mergers, improving the results of individual departments is a key factor in maintaining the profitability of hospitals in the future regarding hernia repair without losing the scope of teaching procedures.

## 1. Introduction

Ventral hernia repair for both primary ventral hernias [1] and incisional hernias is a frequently performed operation [2]. According to the European Hernia Society (EHS), primary ventral hernias are defined as any defects located in the middle line in the area of the umbilicus [3]. According to the EHS, primary epigastric hernias are defined as hernias located in the midline above the umbilicus up to the xiphoid [3]. These hernias were divided into three groups according to their size, Small (0–1 cm), Medium (1–4 cm), and Large (over 4 cm) [3]. In addition, there is no separate classification system for Spiegel’s hernias. Therefore, it is recommended to use the same system as ventral hernias, dividing hernias into three groups according to their size, Small (0–1 cm), Medium (1–4 cm), and Large (over 4 cm) [4]. The classification of incisional hernias, on the other hand, is more complex. A description of the subtypes was made by the EHS [5], and was organized according to the location in the midline from M1 to M5, extending from the xiphoid (proximal) to the os pubis (distal) [5]. These subtypes were defined as, M1 is subxiphoidal (xiphoid up to 3 cm caudal), M2 is epigastric (from 3 cm below the xiphoid is 3 cm above the umbilicus), M3 is umbilical (from 3 cm above to 3 cm below the umbilicus), M4 is considered infraumbilical (from 3 cm below the navel to 3 cm cranial to the os pubis), and lastly M5 is suprapubic (from the os pubis to 3 cm above it) [5]. Analogous to the classification of incisional hernias in the midline, the letters L1–L4 were chosen for lateral hernias. These subtypes were described as, L1 as subcostal, L2 is at the flank, L3 at the iliac, and L4 is lumbar [5]. Since lateral hernias are not discussed in this publication, the description of this classification has been greatly reduced. Furthermore, for both medial and lateral incisional hernias, the size is given in width W1–W3 to distinguish it from primary ventral hernias (W1: <4 cm; W2 ≥ 4–10 cm; W3 ≥ 10 cm) [5]. Moreover, the length of the hernia and whether it is a recurrent hernia can be provided as additional information [5]. Various techniques are available for the treatment of ventral hernias which can be divided into two main groups, open ventral hernia repair, and laparoscopic/endoscopic ventral hernia repair. According to the guidelines of the European and American Hernia Society, for the repair of primary ventral hernias of up to 4 cm, an open surgical technique with the placement of a flat non-absorbable mesh in the preperitoneal layer with edge overlapping of at least 3 cm is recommended [3]. Laparoscopic/endoscopic techniques may be considered as an alternative for larger defects, in obese patients, and in patients with an increased risk of wound infections [3,6]. According to the Guidelines for the Laparoscopic Treatment of Primary Ventral and Ventral Incisional Hernias of the International Endohernia Society (IEHS), the indications for laparoscopic/endoscopic repair continue to increase with further technical progress [6].

Compared to primary ventral hernias, ventral incisional hernias are much more heterogeneous and varied as they range from small to huge defects. Moreover, a loss of abdominal wall integrity may have already occurred, so simple closure without component separation or plastic covering is no longer conceivable [7]. Recently, component separation through endoscopic or endoscopic assisted procedures have been increasingly described [8]. However, they remain reserved only for specialists and specialized centers [8].

Incisional hernia repair requires mesh insertion [8]. In a review [8], the comparison between different surgical techniques does not show any noteworthy differences in terms of reoperation rate, complications, and recurrence [8]. Whether an open or endoscopic approach is chosen [8] and in which position the mesh is inserted (sublay or IPOM) [8], as well as whether a reconstruction of the abdominal wall is feasible and/or necessary, must be decided by the surgeon on a case-by-case basis and based on patient characteristics [8]. However, there is no one unequivocal guideline for primary ventral hernia repair, only general recommendations for certain situations [8].

**Literature Review.** In a metanalysis, the incidence of incisional hernias was described as 4–10% [9], and the incidence for primary ventral hernias was provided as 3–5% [10]. Therefore, procedures for the treatment of both hernia types are regularly performed. As already described above, primary ventral and incisional hernias, as well as their treatment, substantially differ [6,11,12,13,14,15,16]. Several studies have shown that the outcome and the relevant factors of surgical repair for both primary and incisional hernias should be analyzed separately because of different influencing factors that do not correlate between hernia types [6,11,12,13,14,15,16].

In various studies the influence of individual factors on cost and returns was investigated. Factors such as age and sex [3,17], as well as preoperative epidemiological factors and comorbidities (BMI, diabetes mellitus, alcohol consumption, nicotine consumption) [3,17] and postoperative comorbidities (local infections, ASA classification) [3,17] were inspected thoroughly. In this context, the main factors affecting returns were defined as postoperative complications such as wound infections [3], recurrences [3], as well as all conditions that prolong hospital stay and entail additional treatment [3,17]. The complications, in turn, are significantly related to epidemiological factors and comorbidities [3,17]. A BMI of more than 40, ASA class IV, insulin-dependent diabetes mellitus, and deep vein thrombosis were defined as significant factors [3,17]. A BMI of more than 40, unhealthy alcohol consumption, nicotine use, and insulin-dependent diabetes mellitus, as well as all factors that promote wound infection, were described as the most important preoperative aspects, these risks add up and increase the probability of a wound infection and thus a negative financial result. [3,17]. Even though the impact of epidemiological factors on costs is indisputable, Swiss hospitals receive public service contracts, which makes patient selection according to these elements illegal. Even our hospital, a private hospital with a public service mandate, is not allowed to select patients. Therefore, unlike previous studies, we wanted to demonstrate that it is possible to increase the yield of ventral hernia care in the Swiss DRG system independently of comorbidities through process optimization and quality improvement.

**Development of hypothesis.** In an analysis we previously published [18], we were able to demonstrate that it is possible to increase the yield of inguinal hernia care in the SwissDRG system independently of comorbidities through process optimization and quality improvement [18]. Surgical operating time, the total anesthesia time, the number of surgeons present, the insurance state of patients, and surgical technique were identified as vital factors that can increase returns [18].

After reviewing the existing literature [3,6,11,12,13,14,15,16,17,18,19,20] and following our previous publication [18], our hypothesis states that primary ventral hernias and incisional hernias must be analyzed separately, and that ventral hernia repair in the SwissDRg system is deficient, but that an increase in yield is possible [18]. We postulated that individual operations suitable for teaching could be identified. For this purpose, predictive models should be developed using the available data. Our hypothesis further states that, to increase yield, the processes in ventral hernia care must be individual elements that can be defined independently of epidemiological factors and comorbidities, which can lead to an improvement of financial earnings in the SwissDRG system through process optimization and quality improvement. This assumption is supported by the existing literature [3,19,20]. Furthermore, we postulate that the mesh’s price that is used to seal the abdominal wall hernia drives up the costs. In addition, a predictive model will be developed to determine which combination of factors could lead to a financial loss and which countermeasures may be employed to minimize losses. The combination of all the measures mentioned above for preoperative optimization [3,17,19] including the predictive algorithm, process optimization, and quality improvement [3,18,19,20] could be used to increase yields significantly. The literature reviewed confirms this hypothesis [3,17,18,19,20]. For this purpose, we analyzed primary ventral and incisional hernias separately as described above [6,11,12,13,14,15,16] based on their contribution margins (in this case CM4) [21]. In the multi-level contribution calculation (in this case, 4-stages), all costs are deducted from the original operating income gradually (in this case in four steps) so that the contribution margin (in this case CM4) remains at the end [21,22,23]. Since we wanted to analyze the factors that influence this calculated operating result, we have decided to use the contribution margin (CM4) for the calculation.

The already high financial pressure on hospitals is increasing further with the reduction of the base rates in the SwissDRG system. Many hospitals will no longer be able to operate profitably without new concepts. The aim of our analysis is to show a way to improve results in the field of ventral hernia repair. Costs should be visualized and reduced to optimize revenues in the surgical department. At the same time, high efficiency with high quality is essential for optimization and revenue improvement. Furthermore, our analysis should help to identify suitable training operations in the area of ventral hernia repair.

## 2. Materials and Methods

**Data Collection.** Data from all the patients admitted to Wetzikon hospital between January and December 2019 for ventral hernia repair were included in this study. Wetzikon hospital is a private hospital with a public service contract.

**Statistical Analysis.** Linear regression models were constructed for both types of hernias and an overall model was also made. The dependent variable was “Pay” (contribution margin 4 i.e., result after deduction of costs from revenue) [21,22,23] due to the independent variable Hernia (Primary Ventral or Incisional), Male (gender), Age (age), BMI (5 gradations), ASA, Mesh (used/not used), Mesh-Price (the price of the net), Mesh-Size (size of the mesh), Cost of Care (cost of care), Medical Expenses (medical expenses), Experience (experience of the treating physician in four stages), Teaching (whether it was a lesson op), OP Time (OP time) and Anesthesia Time (anesthesia time). Variables are explained in Table 1. To reduce the number of predictors, the costs of the doctor and care were summarized (Cost). By looking at the outlierTest function within the statistical software R, observed outliers were removed. Using the qqPlot and the variance inflation factor (VIF) within R, normality along the values can be observed and multicollinearity was not observable between variables (values < 5), see Table 2.

**Definitions.** The insurance status was either basic care or semi-private and private care. Differences between these statuses are the coverage of extra services, which means the patient is entitled to a double bedroom (semi-private) or a single bedroom (private) and can choose their treating physician [18]. We obtained the variables from the controlling department’s internal data processing system and correlated these with the contribution margin (CM4) [18,21] of individual procedures. The CM4 value indicates a possible over or under-coverage relating to case-specific costs [18,21]. To achieve the base price of a DRG case-based lump sum we multiplied the respective evaluation ratio by the base case rate [18,24].

## 3. Results

Data from 86 consecutive patients was collected (Table 3). The average age of the patients was 56 years old (range: 24–92 years old). In total, 47% of the patients were women and 53% were men. A total 66 primary ventral hernias and 20 incisional hernias were treated. All primary ventral and incisional hernias were repaired in open procedures.

The mean operating time was 62 min (range: 18–201 min) for primary ventral hernias and 70 min (range: 16–111 min) for incisional hernias. The mean anesthesia time was 66 min (range: 30–126 min) for primary ventral hernias, as for incisional hernias, the mean anesthesia time was 74 min (range: 52–94 min). The 86 surgeries were additionally broken down in terms of insurance status and profitability. A mean deficit of −378.17 CHF per case (range: −10,481 to +4444 CHF) occurred across all insurance classes in the Primary Ventral Hernia Group and a mean deficit of −1442.50 CHF per case (range: −5539 to +2739 CHF) in the Incisional Hernia Group. This confirms the first part of our hypothesis that ventral hernia repair is overall a deficient treatment and reflects the existing literature.

Ventral hernia repairs in the scope of OKP (general health insurance) were unprofitable, with a contribution margin of −1147 CHF per case. Hernia interventions in the scope of VVG (private and semi-private insurance) were profitable, with a contribution margin of 2040 CHF per case. Between the two hernia groups, the Primary Ventral Hernia repairs in the scope of OKP (general health insurance) were unprofitable, with a contribution margin of −529 CHF per case. However, in the scope of VVG (private and semi-private insurance) they were profitable, with a contribution margin of 1926 CHF per case. Incisional Hernia repairs in the scope of OKP (general health insurance) were unprofitable, with a contribution margin of −2665 CHF per case. Incisional Hernia interventions in the scope of VVG (private and semi-private insurance) were profitable, with a contribution margin of 2669 CHF per case. The finding that, compared to the overall deficient care of ventral hernias in the scope of Swiss-DRG, patients in the scope of OKP (general health insurance) are treated with a financial deficit, while the care in the scope of VVG (private and semi-private insurance) can be performed profitably is an expected but new result without comparative data in the literature.

### 3.1. Overall Model

**Question.** An overall model was developed to answer the question of what the differences are in treatment costs between the two hernia types in different situations. Diverse circumstances such as a patient’s physical condition—whether it is poor or excellent—variable experience of different surgeons and contrasting mesh prices (unresolving Flat Mesh 102.75 CHF, unresolving Flat Self Gripping Mesh 341.63 CHF, unresolving Flat Intraperitoneal Onlay Mesh 671.27 CHF).

**Model quality.** After four individual cases were removed, which could not be explained well by any model, an overall model with a very good fit of r^2^ = 0.9632 (corrected = 0.9362) was obtained. The prerequisites for a valid regression model such as homoscedasticity and normal distribution of the residuals are satisfied (Non-constant Variance Score Test: = 0.289, Shapiro-Wilk test. = 0.884). The overall model is sufficient to make statements about influencing variables between the two hernia species (Table 4).

**Main influencing factors.** Of all the predictors, the operation time (= −0.403) has the most significant influence on costs, followed by the price of the mesh (= −0.244) with comparably high impact (if the costs of the mesh are increased by s = 309.80 CHF, the total costs will increase 0.25 times to reach = 2560.41 CHF, i.e., 639.82 CHF). The other influencing factors play a rather inferior role.

**Interactions.** In the following section, the main effects of different regression models in both types of hernia are displayed. The numerous interactions between the other influencing factors would certainly be compelling to explore. However, the sample size is not sufficient to model the interactions of two influencing variables for both hernia types in the same model and to quantitatively compare them with each other with the help of higher-order interactions. As already described in the introduction, a joint analysis is not recommended in the literature [6,11,12,13,14,15,16], which we confirm hereby.

**Predictions.** These were created for different situations. The mean values were used for variables that are not mentioned (e.g., operating time). The results of the two hernia types are compared for different situations (Table 5).

For this reason, individual models were created for the two hernia types to investigate different interactions. In all of the following figures, the predictors are mean-centered, that is, zero represents the mean; Figure 1, Figure 2, Figure 3 and Figure 4.

These results reflect the heterogeneity of the two hernia groups (Primary Ventral and Incisional). The result confirms the need for separate analyses for the two types of hernias, and it is consistent with the statements we found in the literature [6,11,12,13,14,15,16].

### 3.2. Incisional Hernia

**Question.** A single model was developed to answer the question of which factors significantly influence the cost of treatment in incisional hernia and to make outcome predictions in certain situations. Table 6 shows the descriptive statistics of the incisional hernias.

**Model quality.** The result was a model with a very good fit of r^2^ = 0.9878 (corrected = 0.9668). The prerequisites for a valid regression model such as homoscedasticity and normal distribution of the residuals are fulfilled (Non constant Variance Score test: =0.488; Shapiro-Wilk test: =0.299). However, due to the small sample size, this single model is only partially suitable to make assertions on all influencing variables for the result of the treatment of incisional hernia. Only a handful of select predictors can be used, op time and costs could not be taken into account. A summary for the model of incisional hernia is displayed in Table 7.

**Main influencing factors.** Of all the predictors, the experience of the surgeon (=0.942, <0.001) has the greatest impact, followed by the cost of the mesh (=−0.500, <0.001) and multiple patients’ characteristics such as BMI (=−0.590, <0.001), gender (=−0.113, <0.055) and age (=−0.026, <0.050), which is consistent with our hypothesis and the literature reviewed [3], see also Table 8.

**Interactions.** In the following section, different regression models of the interactions between various predictors are illustrated, Figure 5 and Figure 6.

**Predictions.** The prediction model shows different scenarios in regard to varied mesh prices, experience, gender, as well as BMI and gives a prediction regarding the returns (Table 9). For variables that are not mentioned (OP time, etc.), their mean values were used. The variable Experience is coded with a number from 1 to 4 (4 = Resident Surgeon, 3 = Attending Surgeon, 2 = chief of service, 1 = chief of Surgery).

The model shows the influence of the experience of the surgeon and BMI. With high experience and low BMI, significant returns are more likely to be achieved or the losses are lower and vice versa, the losses are greater with little experience and high BMI.

### 3.3. Primary Ventral Hernia

**Question.** A single model was developed to answer the question of which factors have a significant effect on the cost of treatment in primary ventral hernias and to determine predictions for the outcome in certain situations. Table 10 shows the descriptive statistics of the primary ventral hernias.

**Model quality.** The result was a model with a very good fit of r^2^ = 0.9523 (corrected = 0.9244). The prerequisites for a valid regression model such as homoscedasticity and normal distribution of the residuals are realized (Non-constant Variance Score Test: = 0.750; Shapiro-Wilk test: = 0.621). Due to the larger sample size, the single model is capable of making statements about influencing variables for the result of the treatment of incisional hernia. However, the number of cases should be higher, because in many figures it becomes clear that the regression line is based only on a few points. A summary for the model of primary ventral hernias is displayed in Table 11.

**Main influencing factors**. Of all the predictors, the operating time (β = 0.564, *p* < 0.001) has the most substantial influence, followed by the costs of the mesh (β = −0.215, *p* < 0.001) and two patient characteristics which are gender (β = 0.143, *p* < 0.01) and BMI (β = −0.087, *p* = 0.074), see Table 12. These results are consistent with our hypothesis and the literature reviewed [4].

**Interactions**. The different regression models of the various interactions between the predictors are shown below. In all of the following figures, the predictors are mean-centered; H. Zero represents the mean; Figure 7, Figure 8, Figure 9 and Figure 10.

**Predictions**. The prediction model shows the yield in various situations regarding various mesh prices, experience, gender, and BMI and gives predications on the yield, Table 13. For variables that are not mentioned (OP time etc.), their mean values were utilized. The variable Experience is coded with a number from 1 to 4 (4 = Resident Surgeon, 3 = Attending Surgeon, 2 = Chief of service, 1 = Chief of Surgery).

From this predictive model, it can be seen that the surgeon’s experience does not influence the result. The most relevant factor, the operating time, was set to a constant value in this model. The model clearly displays the impact of the mesh price and the patient’s BMI on earnings.

## 4. Discussion

The 86 cases analyzed in 2019 showed an average deficit of −378.17 CHF per case (range: −10,481 to +4444 CHF) which occurred across all insurance classes in the Primary ventral Hernia Group and a mean deficit of −1442.50 CHF per case (range: −5539 CHF to +2739) in the Incisional Hernia Group. With a total deficit for the hospital of −70,076 CHF. Our results clearly show that far-reaching measures are necessary to make ventral hernia repair profitable in the OKP system (general health insurance) (CM4 −1147 CHF per case). Regarding the VVG system (private and semi-private insurance), ventral hernia repair was profitable (CM4 of 2040 CHF per case). By implementing measures to increase quality and optimize processes, this yield can also be increased [4]. This clear difference can be explained by the structure of the Swiss health system. This system of the Swiss DRG was introduced in 2012 [25], with far-reaching effects on the OKP scheme. In the Swiss DRG system, hospitals’ income is reduced to flat rates which are calculated according to defined formulas [25]. In Switzerland, there is basic insurance within the scope of OKP. In addition, there are semi-private (SP) and private supplementary insurances (P) within the scope of VVG, the benefits of which are not limited. This can lead to significantly higher income for hospitals compared to general insurance in the OKP system. The weight of the costs that has been increasing for years under this regime has now become even more debilitating due to the COVID-19 pandemic [26]. An additional cost factor is the last-minute introduction of a policy requiring hospitals to increase the number of operations performed on an outpatient basis [P]. This led to considerable losses of several hundred million CHF, which were and still are lacking for the hospitals at a short notice and without prior indication. The only way for hospitals to survive under these enormous losses is to increase quality and economic efficiency. Patient-centered approaches, such as the introduction of Patient-Reported Outcome Measures (PROM), should also be implemented [27,28]. With these viewpoints and with the knowledge of the deficit result (−70,076 CHF) in the hernia repair we have analyzed the performed operations.

Ventral hernia repair with the two subtypes of primary ventral and incisional hernias is a very complex topic with very various subgroups [6,11,12,13,14,15,16]. The incidence of primary ventral hernias is provided as 3–5% [10] and the incidence of incisional hernias is given as 4–10% [9]. The complication rate for primary ventral hernia care is displayed as 3–23% [29,30,31,32]. The rate of complications in incisional hernia care is presented as 12.7–29.1% [33,34,35]. These data show that ventral hernia repair is an important surgical issue. In line with the literature available, our analysis of the overall model has shown that the data are very diverse [6,11,12,13,14,15,16]. We were able to illustrate that of all the predictors in the joint analysis, the operating time (β = −0.403) had the greatest impact on costs, followed by the prices of the mesh (β = −0.250), the anesthesia time (β = −0.248) which contains the operating time as a subset, and the doctor and nursing costs (β = −0.244) with comparably high effect. When comparing the different sub-groups, it could be observed that the influence of the operating time on the result differs substantially and significantly in the two hernia populations (β = −0.269, *p* < 0.001), the same applies to the anesthesia time (β = −0.259, *p* < 0.001). The influence of the patient’s physical condition on the result, regardless of age (ASA class), differed significantly in the hernia groups (β = −0.131, *p* < 0.050). The influence of the patient’s age (regardless of the ASA class) on the result diverged only at the 10% level in the hernia groups, β = 0.089, *p* < 0.100. Given these major differences, the sample size was insufficient to model the interactions of two influencing variables for both types of hernia in the same model and to compare them quantitatively with the help of higher-order interactions. For this reason, we carried out a separate analysis for each subgroup. For the primary ventral hernias, the surgery time (β = 0.564, *p* < 0.001) had the greatest influence, followed by the costs of the mesh (β = −0.215, *p* < 0.001). A nonmodifiable epidemiological factor like Gender (β = −0.143, *p* < 0.01), as well as a modifiable epidemiological factor which is BMI, are also impactful (β = −0.087, *p* = 0.074). This conclusion fits the literature we reviewed in which technical factors, epidemiological factors that can and cannot be modified, and comorbidities are described as significant factors [3,17]. Unlike previously published papers, we also analyzed the interactions between these factors. We were able to illustrate that the impact of the total personnel costs (doctor and care) on the result depends significantly on the ASA level, β = −0.480, *p* < 0.001. This fits with the reviewed literature [3,17] and is explained by the longer stays and the increasing comorbidities [3,17]. To modify these costs, in agreement with the literature, we recommend preoperative internal optimization of this patient insurance [3,17]. In the available data, the influence of the mesh price on the result also depends on the patient’s ASA value (β = −0.343, *p* < 0.001). This underlines the influence of comorbidities on the financial outcome that we have described and corresponds with the literature that preoperative internal optimization would improve the earnings here too [3,17]. The influence of the total personnel costs (doctor and care) on the result depends on the mesh price (β = −0.331, *p* < 0.001). This clearly shows that the mesh price has a significantly greater impact on the result and that only minor improvements can be achieved through measures that affect personnel costs. The influence of the surgery time on the result depends on the mesh price (β = −0.271, *p* < 0.001). This explains the limited potential of cost saving through shortened surgery time when using an expensive mesh. Our predictive model for primary ventral repair gives an indication that there is no significant difference in earnings in regard to different operator experience with modifiable and nonmodifiable epidemiological factors, as well as the same operation time (benchmark) for primary hernia repair. This would mean that this subgroup is very suitable for teaching young surgeons if attention is paid to maintaining quality (operating time). There is an English publication that confirms these findings and even recommends using these operations for teaching in an outpatient setting [36].

At the same time, studies show that teaching operations lead to a longer operating time [36,37]. This problem needs to be compensated for by increasing the quality of teaching and process optimization to enable teaching within a benchmark (operating time). Primary ventral hernia care in outpatient setting would make teaching easier [36], but within the scope of the mandatory health care insurance (OKP) and the Insurance Contract Act (VVG), billing in the outpatient Tarmed Tariff instead of the inpatient SwissDRG system would lead to considerable losses in income (1.9–3.2 times) [38]. Thus, the problem of the decrease in quality of training through lower attendance, due to compliance with the Working Hours Act (swiss law) [39] as well as decreasing numbers of teaching operations due to increasing bureaucracy, remains a current, yet unresolved issue.

However, the incisional hernia subgroup, shows that the surgeon’s experience had the most significant effect (β = 0.942, *p* < 0.001), and not the operating time as with primary ventral hernias. These clear differences coincide with the literature. The treatment of incisional hernias is conducted by specialists and specialized centers [4]. The second biggest influence is the prices of the mesh (β = −0.500, *p* < 0.001). This fits in with the increased complexity of these hernias and their care as described in the literature, as well as the increased complication rate compared to primary ventral hernias [4,33,34,35,40,41]. The epidemiological factor BMI (β = −0.590, *p* < 0.001), as well as non-modifiable epidemiological factors such as gender (β = −0.113, *p* = 0.055) and age (β = −0.026) also have a major influence (*p* < 0.050). This is in accordance with the literature [4,15,17]. The interactions between the predictors illuminated that the operator’s experience (β = −0.374, *p* < 0.050) and the patient’s BMI had a greater impact on the overall result than mesh prices (β = −0.319, *p* < 0.001). This result also fits the existing literature [4,17,41]. Our predictive model was able to show that with constant epidemiological factors and operation time, the financial result decreases in relation to a lower level of experience of the surgeon. This shows that these operations are not suitable for teaching young surgeons when increasing quality and process optimization is desired. In summary, our results confirm both our hypothesis and the results of our literature review.

Relevant epidemiological in primary ventral hernia repair like gender and age cannot be changed but acknowledging them and other relevant epidemiological factors that can be altered (BMI, ASA), is vital for correct planning of the procedure [41].

The pressure to comply with quality standards will continue to increase in the future, and PROM will also gain importance in the future. In the future, it should be possible to compare revenue increasing measures with PROM in a predictive model to illuminate not only the effect on earnings but also the benefits on patients’ health, to choose the optimal preoperative internal optimization, correct surgical procedure, and select a surgeon with the right level of experience [42]. The quality of surgical training is suffering from increasing financial losses [37]. Therefore, we think that the identification of resources which are used for training are of great importance to counteract this problem. Our prediction models can provide a clue to identify suitable and unsuitable operations for teaching and thus avoid quality and profit losses.

One of the limitations of our study was a small sample size with 86 patients. The samples were too small for further subgroup analysis regarding the position and size of the hernias. Furthermore, all operations were open procedures so that it was not possible to compare open and laparoscopic techniques. In the future, studies with larger sample size can conduct subgroup analysis and build predictive models that include pre-qualification, the significant factors we found, and PROM. It is essential to have prediction models about increasing revenue and teaching possibilities, with simultaneous assertions on the benefit for patients’ health.

Furthermore, all operations we were focusing on were open procedures, therefore it was not possible to compare open and laparoscopic techniques that could have a relevant influence on the identified costs. Another limitation is the lack of a follow-up, of comorbidity indices and missing PROM in our analysis. Nevertheless, our analysis revealed significant factors influencing the cost effectiveness of ventral hernia repair. Future prospective multi-centric studies with higher number of cases, subgroup analyses, and an additional comparison of the financial outcome and the PROM outcome are likely to overcome these shortcomings and would be beneficial for our hospital landscape.

## 5. Conclusions

Ventral hernia repair is a deficit-causing treatment under the OKP system (−1147 CHF per case, and a total deficit of −70,076 CHF per year). Our results clearly demonstrate that far-reaching measures to increase earnings are essential in ventral hernia repair under the OKP system (general health insurance). In the scope of the VVG scheme (private and semi-private insurance), the procedure was profitable (CM4 of 2040 CHF per case).

Primary ventral hernias must be considered separately from the incisional hernias [6,11,12,13,14,15,16,41]. The measures that must be taken to increase revenue and quality differ and must be applied differently. The same applies to choosing the appropriate surgical technique and surgeon. Predictive models and knowledge of the patient-specific epidemiological factors can help to make this choice. For primary ventral hernias, preoperative internal optimization of patients’ factors (ASA, BMI) should be considered if feasible. The experience of the surgeon plays a secondary role, which is why primary ventral hernias are suitable for teaching young surgeons [36]. The processes in theatre should be optimized to keep operating time low, which would help to make primary ventral hernia repair a teaching operation without loss of earnings. However, this represents a major challenge for the teaching surgeon. Outpatient care for primary ventral hernias would make teaching easier without any loss of income [36], but in the Swiss healthcare system, within the scope of mandatory health insurance (OKP) billing in the outpatient Tarmed Tariff leads to considerable loss of income (1.9–3.2 times) [38]. Thus, the problem of decreasing training quality is due to lower attendance in compliance with the working time law and decreasing number of teaching operations and increasing bureaucracy.

A solution for this dilemma must be found since the reoperation rate decreases with increasing surgeon experience [42]. This directly affects the PROM, as well as the quality of treatment. Therefore, even in this challenging situation, high-quality training must be enforced, which could be ensured by increasing the efforts of the training surgeons.

In the case of incisional hernias, it has been shown that the surgeon’s experience is the decisive factor for the increase in yield and quality. Therefore, these procedures are of limited use for teaching young surgeons. Unlike primary ventral hernias, the cost of the mesh is the second most important factor here. Preoperative internal optimization (BMI) can also reduce complications and save costs in incisional hernias. Some relevant epidemiological factors in primary ventral hernia repair (such as gender and age) cannot be altered, but understanding the epidemiological factors and the difference to the primary ventral hernias is important for correct planning of the procedure [41].

The pressure to comply with quality standards will continue to increase in the future [40], and PROM will also gain importance. In a predictive model, the measures designed to increase the yield should be compared to the PROM, to determine the benefits for the health of the patient in addition to the effect on earnings and to select the optimal preoperative internal optimization, surgical procedure, and the surgeon with the right level of experience to increase revenue with maximal benefit for the health of the patient at the same time [42].

Since the already high financial pressure on hospitals is increasing further with the reduction of the base rates in the SwissDRG system [43] many hospitals will not be able to operate profitably without new concepts [43]. In addition to purchasing management, new construction projects, and mergers, improving the results of individual departments is a key factor in maintaining the profitability of hospitals in the future regarding hernia repair. Our analysis shows a way to improve outcomes in the area of ventral hernia repair. Costs can be visualized and reduced to optimize revenue in the surgical department. At the same time, high efficiency with high quality is important for optimization and increased revenue. However, this is opposed to the necessary training of juniors. The decreasing number of training procedures makes it essential to identify suitable operations, which can be used cost effectively for teaching. Our analysis has helped to identify primary ventral hernias as a suitable training procedure with simultaneous cost efficiency.

It is crucial to find prediction models concerned with increasing revenue and teaching possibilities, with simultaneous assertions on the benefit for patients’ health. Therefore, this type of analysis should be extended to other areas of surgery for evaluating profitability and ensuring young surgeons’ education in many fields.

## Figures and Tables

**Figure 1 healthcare-09-01226-f001:**
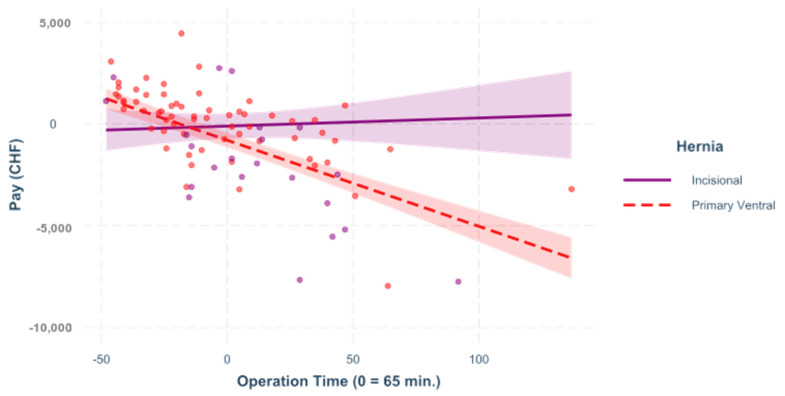
Surgery time. The influence of op time on the result differs scheme = 0.269, < 0.001. This figure shows a regression that was built to see the impact of the operation time on the variable “pay” for incisional and for primary ventral cases.

**Figure 2 healthcare-09-01226-f002:**
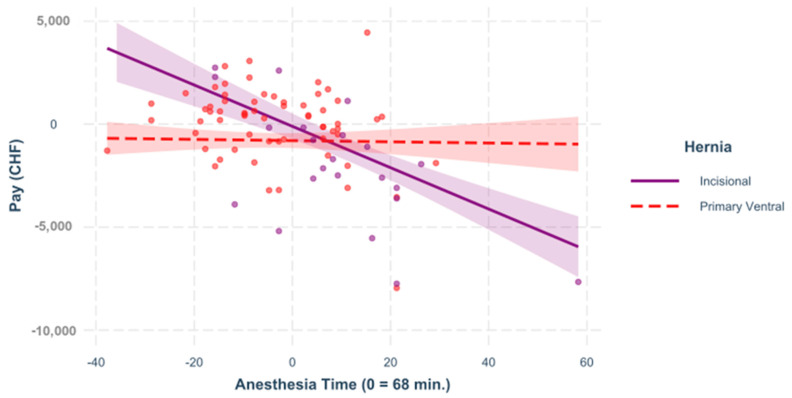
Anesthesia time. The influence of anesthesia time on the result differs substantially and significantly in both hernia populations, =−0.259, <0.001. In this figure, the impact of the anesthesia time on the variable “pay” for incisional and for primary ventral cases is demonstrated by the regression.

**Figure 3 healthcare-09-01226-f003:**
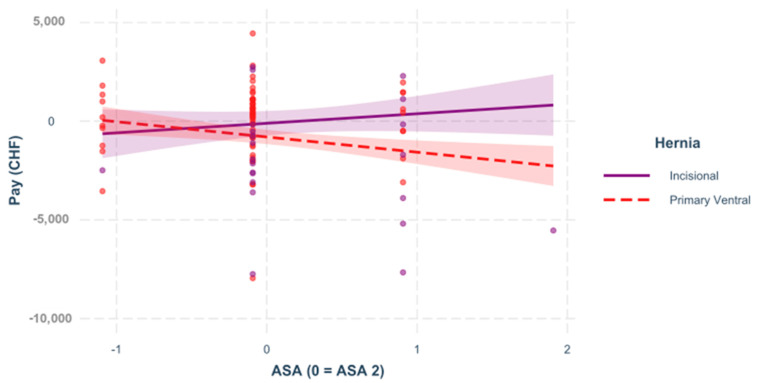
ASA. The influence of the patient’s physical condition (regardless of age) on the result differs significantly in the hernia groups, =−0.131, <0.050. The regression demonstrates the impact of the ASA on the variable “pay” for incisional and for primary ventral cases.

**Figure 4 healthcare-09-01226-f004:**
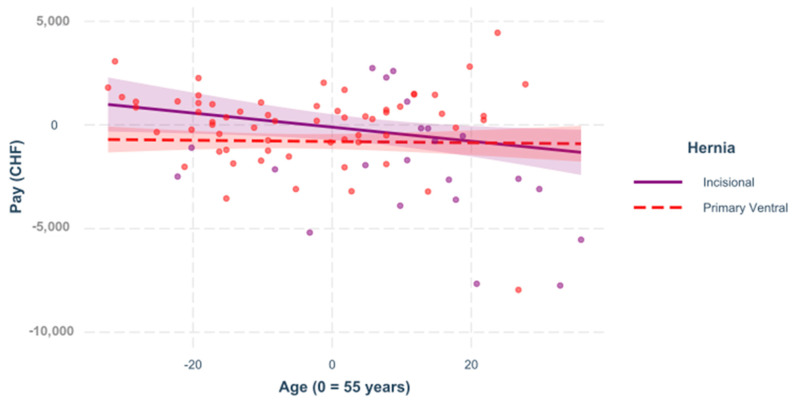
Age. The influence of the age of the patient (regardless of the ASA) on the outcome differs only at the 10% level in the hernia groups, =0.089, <0.100. Here, a regression was built to see the impact of the age on the variable “pay” for incisional and for primary ventral cases.

**Figure 5 healthcare-09-01226-f005:**
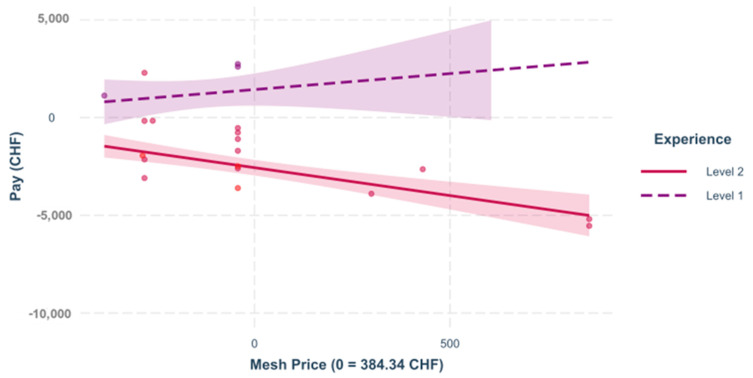
Mesh price and experience. The influence of the mesh price on the result depends significantly on the experience of the surgeon, =−0.374, <0.050. The figure shows the regression that was built to see the impact of the mesh price on the variable “pay” for incisional depending on the experience of the surgeon.

**Figure 6 healthcare-09-01226-f006:**
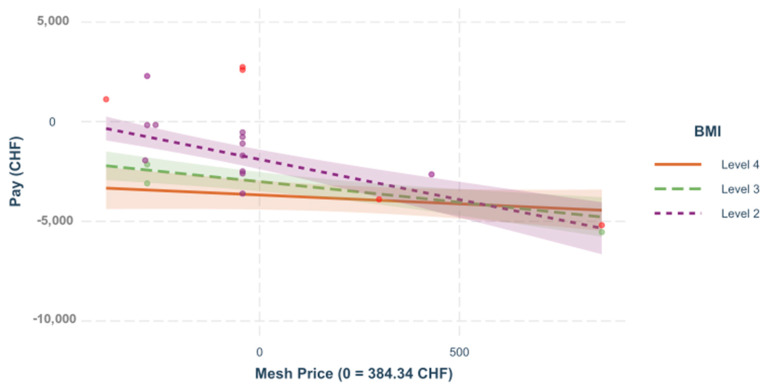
Mesh price and BMI. The influence of mesh price on the result depends significantly on the BMI value of the patient, =−0.319, <0.001. A regression was built to see the impact of the mesh price on the variable “pay” for incisional, depending on the different levels of BMI.

**Figure 7 healthcare-09-01226-f007:**
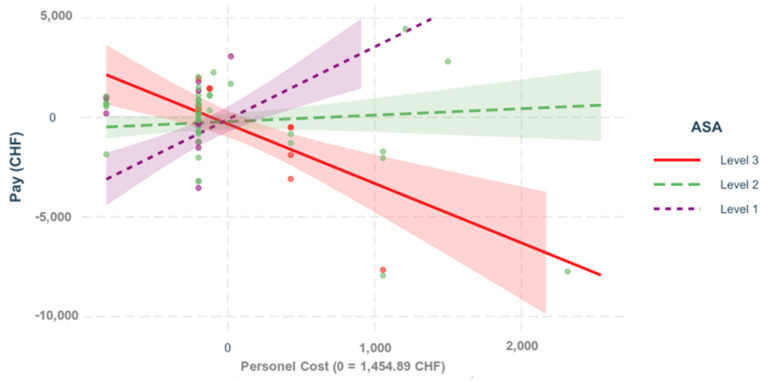
Personnel costs and ASA. The influence of the costs on the result depends on Scheme = 0.480. *p* < 0.001. The figure shows a regression, which reflects the impact of the personnel costs on the variable “pay” for primary ventral cases, depending on the different ASA scores.

**Figure 8 healthcare-09-01226-f008:**
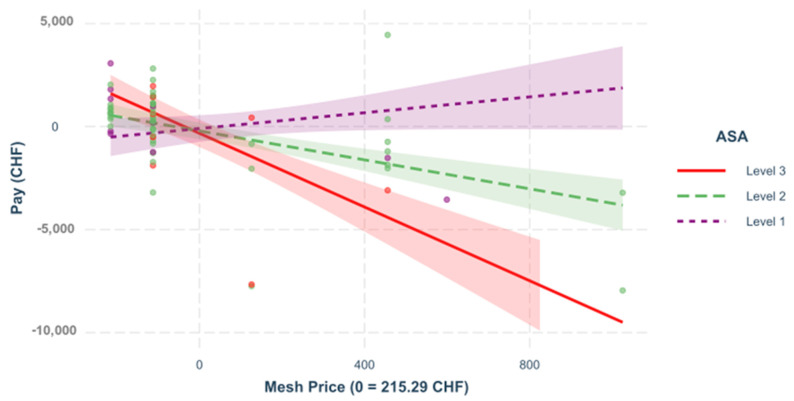
Mesh price and ASA. The influence of the *mesh price* on the result depends on Scheme = 0.343. *p* < 0.001. Here, a regression was built to see the impact of the mesh price on the variable “pay” for primary ventral cases, depending on the different ASA scores.

**Figure 9 healthcare-09-01226-f009:**
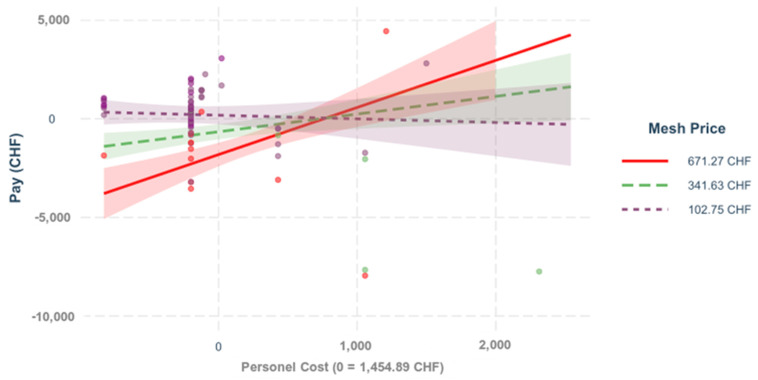
Mesh price and personnel costs. The influence of the costs on the result depends on scheme = 0.331. *p* < 0.001. This regression demonstrates the impact of the personnel costs on the variable “pay” for primary ventral cases, depending on the different mesh prices.

**Figure 10 healthcare-09-01226-f010:**
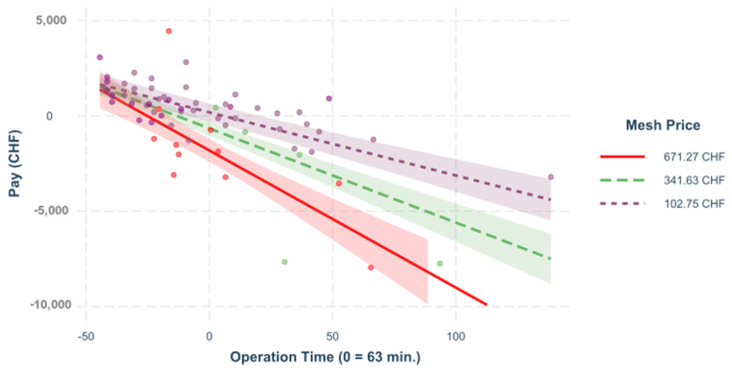
Mesh price and surgery time. The influence of the operating time on the result depends significantly on the mesh price, β = −0.271, *p* < 0.001. A regression was built to see the impact of the operation time on the variable “pay” for primary ventral cases, depending on the different mesh prices.

**Table 1 healthcare-09-01226-t001:** Variables used for statistical analysis and their definition and a formula for their calculation.

Variable	Definition	Formula/Categories
Pay	The amount which was spend for a patient	Sum of all costs
MeshPrice	The price of the mesh used for each patient	
Cost	Sum of the costs	Cost of Care+Cost of medical expenses+mesh price
OPTime	Time spend for an operation (in minutes)	
AnesthesiaTime	Sum of induction time and emergence time as reported by the anesthesiologist (in minutes)	Induction time+emergence time
Experience	Experience of the doctor defined by his/her position within the department	- CSu (Chief of Surgery) - CSe (Chief of Service) - AS (Attending Surgeon)
Age	Patients age in years	Date of operation−Date of birth
BMI	Body Mass Index	Patient weight (kg)/(Patient height (cm))^2^
ASA	American Society of Anesthesiologists	I, II, III, IV, V, VI
MaleYes	Indication of patients gender	1 = male, 0 = female
Hernia primary ventral	Indication of type of hernia	1 = primary ventral, 0 = incisional

**Table 2 healthcare-09-01226-t002:** Variance inflation factor.

Mesh Price	Cost	OPTime	AnesthesiaTime	Experience	Age
3.1	4.9	4.9	3.5	2.6	2.1
**BMI**	**ASA**	**Male**	**Hernia**		
1.9	2.2	2.9	3.7		

**Table 3 healthcare-09-01226-t003:** Descriptive statistics of the whole sample.

Variables	n	M	SD	Min	Q1	Md	Q3	Max	Range
Pay	86	−625.69	2560.41	−10,481.00	−1827.75	−133.50	974.00	4444.00	14,925.00
Mesh Prize	86	254.61	309.80	0.00	102.75	102.75	341.63	1240.65	1240.65
Cost (med. + care)	86	1583.50	706.85	628.00	1256.00	1256.00	1884.00	3996.00	3368.00
OP Time (min)	86	64.16	33.89	16	40	57	88	201	185
Anesthesia Time (min)	86	67.74	15.48	30	56	66	77	126	96
Experience	86	2.52	0.84	1	2	2	3	4	3
Age	86	56.22	16.79	24	41	58	69	92	68
BMI	86	2.26	1.05	1	1	2	3	5	4
*ASA*	86	2.09	0.61	1	2	2	2	4	3

**Table 4 healthcare-09-01226-t004:** Summary of the overall model.

	Estimate	Std. Error	*t* Value	Pr(>|t|)	
(Intercept)	−108.03	310.29	−0.34	0.72	
MeshPrice	−2.07	0.40	−5.17	<0.001	***
Cost	−0.27	0.24	−1.09	0.27	
OPTime	3.99	7.77	0.51	0.60	
AnesthesiaTime	−100.34	14.29	−7.02	<0.001	***
Experience	57.36	137.06	0.41	0.67	
Age	−33.94	15.19	−2.23	0.03	*
BMI	−7.00	92.42	−0.07	0.93	
ASA	484.03	417.43	1.16	0.25	
MaleYes	415.19	208.14	1.99	0.051	
Hernia primary ventral	−691.90	308.75	−2.24	0.029	*
MeshPrice:Cost	0.001	0.0007	2.12	0.038	*
MeshPrice:OPTime	−0.084	0.012	−6.85	<0.001	***
MeshPrice:AnesthesiaTime	0.085	0.03	2.67	0.01	*
MeshPrice:Age	0.12	0.02	4.55	<0.001	***
MeshPrice:BMI	−1.27	0.32	−3.94	<0.001	***
MeshPrice:ASA	−1.45	0.57	−2.50	0.01	*
Cost:OPTime	−0.018	0.005	−3.38	0.001	**
Cost:AnesthesiaTime	−0.065	0.01	−6.47	<0.001	***
Cost:Experience	−0.34	0.21	−1.58	0.11	
Cost:Age	0.09	0.01	6.24	<0.001	***
Cost:ASA	−3.16	0.36	−8.73	<0.001	***
Cost:MaleYes	−1.16	0.33	−3.48	0.001	**
OPTime:Age	−1.58	0.31	−5.05	<0.001	***
OPTime:ASA	15.47	6.87	2.25	0.02	*
OPTime:Hernia primary ventral	−46.30	8.70	−5.31	<0.001	***
AnesthesiaTime:Experience	13.70	9.02	1.51	0.13	
AnesthesiaTime:ASA	56.86	13.19	4.30	<0.001	***
AnesthesiaTime:MaleYes	−24.73	13.64	−1.81	0.07	
AnesthesiaTime:Hernia primary ventral	97.41	17.31	5.62	<0.001	***
Experience:Age	24.13	8.90	2.70	0.009	**
Experience:BMI	−229.55	109.31	−2.10	0.04	*
Age:ASA	17.06	10.26	1.66	0.1	
Age:Hernia primary ventral	30.95	18.04	1.71	0.09	
BMI:ASA	637.77	181.05	3.52	<0.001	***
ASA:MaleYes	1090.38	354.55	3.07	0.003	**
ASA:Hernia primary ventral	−1255.97	471.89	−2.66	0.01	*

*p* < 0.001 ***, *p* < 0.01 **, *p* < 0.05 *.

**Table 5 healthcare-09-01226-t005:** Predictions in terms of mesh price, experience, gender, ASA, hernia and pay. In this table the price which needs to be paid is predicted by using different inputs which are plugged into the R function “predict” for model predictions.

MeshPrice	Experience	Male	ASA	Hernia	Pay
102.75	1	No	1	Incisional	−584.36
102.75	1	No	1	Primary ventral	−20.29
102.75	1	No	2	Incisional	119.55
102.75	1	No	2	Primary ventral	−572.36
102.75	1	No	3	Incisional	823.45
102.75	1	No	3	Primary ventral	−1124.42
102.75	1	Yes	1	Incisional	−1259.56
102.75	1	Yes	1	Primary ventral	−695.49
102.75	1	Yes	2	Incisional	534.73
102.75	1	Yes	2	Primary ventral	−157.17
102.75	1	Yes	3	Incisional	2329.02
102.75	1	Yes	3	Primary ventral	381.16
102.75	4	No	1	Incisional	−412.26
102.75	4	No	1	Primary ventral	151.80
102.75	4	No	2	Incisional	291.64
102.75	4	No	2	Primary ventral	−400.26
102.75	4	No	3	Incisional	995.54
102.75	4	No	3	Primary ventral	−952.33
102.75	4	Yes	1	Incisional	−1087.46
102.75	4	Yes	1	Primary ventral	−523.40
102.75	4	Yes	2	Incisional	706.83
102.75	4	Yes	2	Primary ventral	14.92
102.75	4	Yes	3	Incisional	2501.12
102.75	4	Yes	3	Primary ventral	553.25
341.63	1	No	1	Incisional	−731.83
341.63	1	No	1	Primary ventral	−167.77
341.63	1	No	2	Incisional	−373.81
341.63	1	No	2	Primary ventral	−1065.71
341.63	1	No	3	Incisional	−15.78
341.63	1	No	3	Primary ventral	−1963.65
341.63	1	Yes	1	Incisional	−1407.03
341.63	1	Yes	1	Primary ventral	−842.97
341.63	1	Yes	2	Incisional	41.38
341.63	1	Yes	2	Primary ventral	−650.52
341.63	1	Yes	3	Incisional	1489.79
341.63	1	Yes	3	Primary ventral	−458.08
341.63	4	No	1	Incisional	−559.74
341.63	4	No	1	Primary ventral	4.32
341.63	4	No	2	Incisional	−201.71
341.63	4	No	2	Primary ventral	−893.62
341.63	4	No	3	Incisional	156.31
341.63	4	No	3	Primary ventral	−1791.56
341.63	4	Yes	1	Incisional	−1234.94
341.63	4	Yes	1	Primary ventral	−670.88
341.63	4	Yes	2	Incisional	213.47
341.63	4	Yes	2	Primary ventral	−478.43
341.63	4	Yes	3	Incisional	1661.89
341.63	4	Yes	3	Primary ventral	−285.98
671.27	1	No	1	Incisional	−9.34
671.27	1	No	1	Primary ventral	−371.28
671.27	1	No	2	Incisional	−1054.60
671.27	1	No	2	Primary ventral	−1746.51
671.27	1	No	3	Incisional	−1173.87
671.27	1	No	3	Primary ventral	−3121.74
671.27	1	Yes	1	Incisional	−1610.54
671.27	1	Yes	1	Primary ventral	−1046.48
671.27	1	Yes	2	Incisional	−639.42
671.27	1	Yes	2	Primary ventral	−1331.32
671.27	1	Yes	3	Incisional	331.71
671.27	1	Yes	3	Primary ventral	−1616.16
671.27	4	No	1	Incisional	−763.25
671.27	4	No	1	Primary ventral	−199.18
671.27	4	No	2	Incisional	−882.51
671.27	4	No	2	Primary ventral	−1574.42
671.27	4	No	3	Incisional	−1001.78
671.27	4	No	3	Primary ventral	−2949.65
671.27	4	Yes	1	Incisional	−1438.45
671.27	4	Yes	1	Primary ventral	−874.38
671.27	4	Yes	2	Incisional	−467.32
671.27	4	Yes	2	Primary ventral	−1159.23
671.27	4	Yes	3	Incisional	503.80
671.27	4	Yes	3	Primary ventral	−1444.07

**Table 6 healthcare-09-01226-t006:** Descriptive statistics of the sample with incisional hernia.

Variables	n	M	SD	Min	Q1	Md	Q3	Max	Range
Pay	20	−1442.50	2383.39	−5539.00	−2757.00	−1821.00	−170.00	2739.00	8278.00
Mesh Prize	20	384.34	353.35	0.00	102.75	341.63	341.63	1240.65	1240.65
Cost (med.+care)	20	2007.90	715.55	1256.00	1256.00	1884.00	2512.00	3140.00	1884.00
OP Time	20	69.85	27.27	16	50	68	91	111	95
Anesthesia Time	20	73.80	12.20	52	65	75	83	94	42
Experience	20	2.00	0.56	1	2	2	2	3	2
Age	20	65.80	14.64	34	62	67	73	92	58
BMI	20	2.60	1.10	1	2	2	3	5	4
ASA	20	2.35	0.67	1	2	2	3	4	3

**Table 7 healthcare-09-01226-t007:** Summary of the model for incisional hernia.

Coefficients:	Estimate	Std. Error	*t* Value	Pr(>|t|)	
(Intercept)	−2563.00	169.93	−15.08	<0.001	***
MeshPrice	2.85	0.47	−5.96	<0.001	***
Experience	−3994.26	315.71	−12.65	<0.001	***
Age	28.00	10.12	−2.76	0.02	*
BMI	−1114.61	238.08	−4.68	0.002	**
MaleYes	522.73	227.94	2.29	0.05	
MeshPrice:Expe-rience	−4.48	1.45	−3.08	0.017	*
MeshPrice:BMI	1.96	0.33	5.94	<0.001	***
MeshPrice:MaleYes	−1.03	0.67	−1.53	0.16	
Experience:Age	−47.84	29.40	−1.62	0.14	
Experience:BMI	−2157.58	298.23	−7.23	<0.001	***
Age:MaleYes	47.60	26.90	1.77	0.12	
BMI:MaleYes	−339.63	283.38	−1.19	0.26	

*p* < 0.001 ***, *p* < 0.01 **, *p* < 0.05 *.

**Table 8 healthcare-09-01226-t008:** Coefficients of the model for incisional hernia.

	Estimate	Std. Error	*t* Value	Pr(>|t|)	Beta	Tolerances
MeshPrice	−2.9	0.48	−6.0	<0.001	−0.500	0.35
Experience	−3994.3	315.71	−12.7	<0.001	−0.942	0.32
Age	−28.0	10.12	−2.8	0.027	−0.026	0.45
BMI	−1114.6	238.08	−4.7	0.002	−0.590	0.15
MaleYes	522.7	227.95	2.3	0.055	0.113	0.73
MeshPrice:Experience	−4.5	1.45	−3.1	0.017	−0.374	0.38
MeshPrice:BMI	2.0	0.33	5.9	<0.001	0.319	0.32
MeshPrice:Male Yes	−1.0	0.68	−1.5	0.16	−0.079	0.35
Experience:Age	−47.8	29.41	−1.6	0.14	−0.165	0.20
Experience:BMI	−2157.6	298.23	−7.2	<0.001	−0.557	0.41
Age:MaleYes	47.6	26.90	1.8	0.12	0.150	0.15
BMI:MaleYes	−339.6	283.38	−1.2	0.26	−0.080	0.16

**Table 9 healthcare-09-01226-t009:** Predictive model 1 in terms of mesh price, experience, gender, BMI, pay. In this table, the price, which needs to be paid, is predicted by using different inputs, which are plugged into the R function “predict” for model predictions.

Mesh Price	Experience	Male	BMI	Pay
102.75	2	Yes	2	260.69
102.75	2	No	2	−758.56
341.63	2	Yes	2	−951.08
102.75	3	Yes	2	−1175.30
341.63	2	No	2	−1722.00
102.75	2	Yes	3	−1746.51
102.75	3	No	2	−2194.54
102.75	2	No	3	−2426.13
341.63	2	Yes	3	−2489.19
671.27	2	Yes	2	−2623.26
341.63	2	No	3	−2920.47
671.27	2	No	2	−3051.48
341.63	3	Yes	2	−3459.14
671.27	2	Yes	3	−3514.05
671.27	2	No	3	−3602.64
102.75	2	Yes	4	−3753.71
341.63	2	Yes	4	−4027.30
102.75	2	No	4	−4093.69
341.63	2	No	4	−4118.95
671.27	2	No	4	−4153.80
341.63	3	No	2	−4230.05
671.27	2	Yes	4	−4404.84
102.75	3	Yes	3	−5340.08
102.75	3	No	3	−6019.69
671.27	3	Yes	2	−6610.71
671.27	3	No	2	−7038.93
341.63	3	Yes	3	−7154.83
341.63	3	No	3	−7586.11
102.75	3	Yes	4	−9504.86
671.27	3	Yes	3	−9659.08
671.27	3	No	3	−9747.67
102.75	3	No	4	−9844.84
341.63	3	Yes	4	−10,850.52
341.63	3	No	4	−10,942.17
671.27	3	No	4	−12,456.41
671.27	3	Yes	4	−12,707.45

**Table 10 healthcare-09-01226-t010:** Descriptive statistics of the sample with primary ventral hernia.

Variables	n	M	SD	Min	Q1	Md	Q3	Max	Range
Pay	66	−378.17	2577.77	−10,481.00	−1114.25	260.00	1039.50	4444.00	14,925.00
Mesh Prize	66	215.29	286.81	0.00	102.75	102.75	281.91	1240.65	1240.65
Cost (med.+care)	66	1454.89	656.88	628.00	1256.00	1256.00	1352.25	3996.00	3368.00
OP Time	66	62.44	35.66	18	39	52	81	201	183
Anesthesia Time	66	65.91	15.98	30	54	64	75	126	96
Experience	66	2.68	0.84	1	2	2	3	4	3
Age	66	53.32	16.41	24	40	54	64	89	65
BMI	66	2.15	1.03	1	1	2	3	5	4
ASA	66	2.02	0.57	1	2	2	2	3	2

**Table 11 healthcare-09-01226-t011:** Summary of the model for primary ventral hernia.

Coefficients:	Estimate	Std. Error	*t* Value	Pr(>|t|)	
(Intercept)	−213.63	197.51	−1.08	0.28	
MeshPrice	−3.51	0.62	−5.60	<0.001	***
Cost	0.32	0.32	1.00	0.32	
OPTime	−40.75	3.32	−12.24	<0.001	***
AnesthesiaTime	18.18	15.11	1.20	0.23	
BMI	−218.46	119.11	−1.83	0.07	
ASA	−114.65	259.09	−0.44	0.66	
MaleYes	734.69	241.71	3.04	0.004	**
TeachingYes	21.47	230.31	0.09	0.92	
MeshPrice:Cost	0.004	0.001	4.12	<0.001	***
MeshPrice:OPTime	−0.068	0.016	−4.21	<0.001	***
MeshPrice:BMI	−1.047	0.45	−2.31	0.02	*
MeshPrice:ASA	−5.43	0.99	−5.44	<0.001	***
MeshPrice:Male Yes	2.90	1.00	2.88	0.006	**
Cost:OPTime	−0.03	0.005	−6.20	<0.001	***
Cost:AnesthesiaTime	0.04	0.018	2.60	0.012	*
Cost:BMI	0.56	0.20	−2.68	0.010	*
Cost:ASA	−3.31	0.69	−4.80	<0.001	***
Cost:MaleYes	0.94	0.51	1.84	0.07	
OPTime:AnesthesiaTime	−0.86	0.35	−2.43	0.01	*
OPTime:ASA	17.38	7.95	2.18	0.034	*
AnesthesiaTime:MaleYes	−34.44	16.96	−2.03	0.048	*
AnesthesiaTime:Teaching Yes	−27.79	16.07	−1.72	0.091	
BMI:ASA	833.60	220.31	3.78	<0.001	***
ASA:Teaching Yes	−1278.21	554.76	−2.30	0.026	*

*p* < 0.001 ***, *p* < 0.01 **, *p* < 0.05 *.

**Table 12 healthcare-09-01226-t012:** Coefficients of the model for primary ventral hernia.

	Estimate	Std. Error	*t* Value	Pr(>|t|)	Beta	Tolerances
MeshPrice	−3.512	0.626	−5.607	0.000	−0.215	0.239
Cost	0.326	0.325	1.003	0.322	0.215	0.170
OPTime	−40.757	3.329	−12.242	0.000	−0.564	0.548
AnesthesiaTime	18.181	15.117	1.203	0.236	−0.064	0.132
BMI	−218.468	119.114	−1.834	0.074	−0.087	0.517
ASA	−114.655	259.095	−0.443	0.660	−0.123	0.356
MaleYes	734.694	241.712	3.040	0.004	0.143	0.525
TeachingYes	21.475	230.315	0.093	0.926	0.004	0.632
MeshPrice:Cost	0.005	0.001	4.120	0.000	0.331	0.201
MeshPrice:OPTime	−0.068	0.016	−4.214	0.000	−0.271	0.270
MeshPrice:BMI	−1.047	0.452	−2.315	0.026	−0.120	0.481
MeshPrice:ASA	−5.432	0.998	−5.442	0.000	−0.343	0.422
MeshPrice:Male Yes	2.901	1.006	2.884	0.006	0.162	0.242
Cost:OPTime	−0.035	0.006	−6.209	0.000	−0.316	0.239
Cost:AnesthesiaTime	0.049	0.019	2.602	0.013	0.199	0.088
Cost:BMI	−0.563	0.210	−2.680	0.011	−0.147	0.160
Cost:ASA	−3.316	0.690	−4.803	0.000	−0.480	0.116
Cost:MaleYes	0.947	0.514	1.842	0.073	0.121	0.280
OPTime:AnesthesiaTime	−0.860	0.353	−2.439	0.019	−0.190	0.204
OPTime:ASA	17.390	7.958	2.185	0.035	0.137	0.355
AnesthesiaTime:MaleYes	−34.450	16.961	−2.031	0.049	−0.107	0.344
AnesthesiaTime:TeachingYes	−27.793	16.073	−1.729	0.091	−0.083	0.229
BMI:ASA	833.601	220.314	3.784	0.000	0.189	0.381
ASA:TeachingYes	−1278.219	554.768	−2.304	0.026	−0.135	0.273

**Table 13 healthcare-09-01226-t013:** Forecasts model 2 in terms of mesh price, experience, gender, BMI, pay. Here, the price, which is needed to be paid is predicted by using different inputs which are plugged into the R function “predict” for model predictions.

MeshPrice	Experience	Male	BMI	Pay
102.75	2	Yes	2	605.03
102.75	3	Yes	2	605.03
102.75	2	Yes	3	504.44
102.75	3	Yes	3	504.44
341.63	2	Yes	2	497.05
341.63	3	Yes	2	497.05
102.75	2	Yes	4	403.84
102.75	3	Yes	4	403.84
671.27	2	Yes	2	348.03
671.27	3	Yes	2	348.03
102.75	2	No	2	196.84
102.75	3	No	2	196.84
341.63	2	Yes	3	146.26
341.63	3	Yes	3	146.26
102.75	2	No	3	96.25
102.75	3	No	3	96.25
102.75	2	No	4	−4.35
102.75	3	No	4	−4.35
341.63	3	Yes	4	−204.52
341.63	2	Yes	4	−204.52
341.63	2	No	2	−604.16
341.63	3	No	2	−604.16
671.27	2	Yes	3	−347.99
671.27	3	Yes	3	−347.99
341.63	2	No	3	−954.95
341.63	3	No	3	−954.95
671.27	2	Yes	4	−1044.02
671.27	3	Yes	4	−1044.02
341.63	2	No	4	−1305.73
341.63	3	No	4	−1305.73
671.27	2	No	2	−1709.50
671.27	3	No	2	−1709.50
671.27	2	No	3	−2405.53
671.27	3	No	3	−2405.53
671.27	2	No	4	−3101.55
671.27	3	No	4	−3101.55

## Data Availability

The datasets used and/or analyzed during the current study are available from the corresponding author on reasonable request.

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
