# Peer review of "Analysis of Factors Relevant to Revenue Improvement in Ventral Hernia Repair, Their Influence on Surgical Training, and Development of Predictive Models: An Economic Evaluation"

_healthcare, 2021, doi:10.3390/healthcare9091226_

Round 1

Reviewer 1 Report

General comments

The aim of the manuscript is to carry out an economic evaluation of a specific surgical procedure (ventral hernia repair) performed in a Swiss private hospital (with a public service contract). For this purpose, linear regression models are applied to patient data collected in 2019.

Being an obviously suitable matter for the Journal, its presentation uses very technical language, which makes the manuscript difficult to read for those who do not have medical knowledge.

Specific comments

In specific terms, I have some methodological doubts, which I recommend that the authors clarify.

  • First, I think it would have been important to make it clear that there are no multi-collinearity problems between the explanatory variables. As the authors know, a simple correlation analysis between these variables usually allows detecting that type of problem.
  • Secondly, by analyzing the descriptive statistics in tables 4 and 8 -- and viewing some of the figures -- one gets the feeling that there will be some ‘outliers’ in the data. If so, the authors should check to what extent its presence may distort the results of the linear regression models.
  • Thirdly, as a mere suggestion, it would have been interesting to use a general-to-specific econometric methodology, presenting an estimated model in which all the (remaining) explanatory variables can be considered significant, from a statistical point of view.

Author Response

Dear reviewer 1,

we thank you for the insightful comments on our manuscript. I am attaching the point-by-point response to your suggestions. 

Anas Taha

Reviewer 2 Report

In the attached document, you can find my comments.

Author Response

Dear Reviewer 2,

thank you for the insightful comments on our manuscript. Please find attached our response to your comments.

Sincerely,

Anas Taha
